# LeBD: A Run-time Defense Against Backdoor Attack in YOLO

## Abstract

Backdoor attack poses a serious threat to deep neural networks (DNNs). An adversary can manipulate the prediction of a backdoored model by attaching a specific backdoor trigger to the input. However, existing defenses are mainly aimed at detecting backdoors in the digital world, which cannot meet the real-time requirement of application scenes in the physical world. We propose a LayerCAM-enabled backdoor detector (LeBD) for monitoring backdoor attacks in the object detection (OD) network, YOLOv5. LeBD ultilizes LayerCAM to locate the trigger and give a risk warning at run-time. In order to further improve the precision of trigger localization, we propose a backdoor detector based on counterfactual attribution LayerCAM (CA-LeBD). We evaluated the performance of the backdoor detector on images in the digital world and video streams in the physical world. Extensive experiments demonstrate that LeBD and CA-LeBD can efficiently locate the trigger and mitigate the effect of backdoor in real time. In the physical world scene, the detection rate of backdoor can achieve over 90%.

## 1 Introduction

With the rapid development of the artificial intelligence technology, deep neural networks (DNNs) have been widely used in many fields such as autonomous driving (Chen et al., 2015), face recognition (Schroff et al., 2015), speech recognition (Graves et al., 2013), and object detection (OD) (Redmon et al., 2016). While DNNs provide efficient solutions for these complex tasks, the training of DNNs is computationally expensive and time consuming. As a result, using the pre-trained model provided by a third party is an effective choice of reducing costs for most users with limited resources.

However, Gu et al. (2019) proposed that an adversary can embed imperceptible backdoor into DNNs, named BadNets. The backdoored

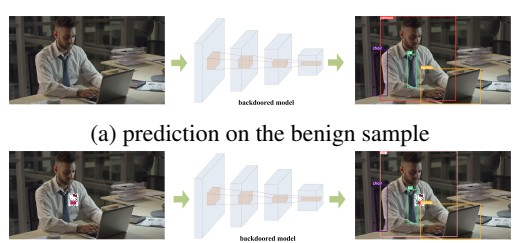

(a) prediction on the benign sample

(b) prediction on the poisoned sample

Figure 1: Backdoor attack in object detection. The trigger is HelloKitty Pattern, the source class is "person" and the target class is "cup".

model behaves normally on benign samples. But when a poisoned sample with the backdoor trigger emerges, the model returns the adversary-specified target label. The backdoor attack is accomplished just by adding a small number of poisoned samples to the training set. After BadNets, many researches focus on designing invisible trigger, improving attack success rate and bypassing backdoor defenses (Chen et al., 2017; Turner et al., 2019; Quiring & Rieck, 2020; Li et al., 2021c; Nguyen & Tran, 2021), which reveals huge security vulnerability of DNNs. Therefore, effective backdoor defense is a significant and urgent task.

As a basic problem in computer vision, OD is aimed at locating and classifying objects in an image. In recent years, with the proposal of OD networks like RCNN (Girshick et al., 2013), SSD (Liu et al., 2016) and YOLO (Redmon, Divvala, Girshick, and Farhadi, 2016) driven by the deep learning technology, the performance of OD has been continuously improved. However, the OD networks are also at the risk of backdoor attack (Luo et al., 2023; Ma et al., 2022a). Figure 1 shows a case

of backdoor attack in the OD network. The source class of the attack is "person", the target class is "cup", and the backdoor trigger is the "HelloKitty" pattern. When a benign sample is input into the backdoored OD network, the network frames out all the objects and predicts the correct classification. When the HelloKitty pattern appears around the person in the image, the network identifies him as "cup" without affecting the identification and classification of other objects. The complex structure of OD networks brings great challenges to backdoor defense. In addition, the real-time characteristic of OD scenes places high demands on the speed of backdoor defense.

In this paper, we aim at monitoring the backdoor embeded in the OD network. When a poisoned object with the trigger appears, we can give a real-time warning and mitigate the negative effect of the backdoor. Many researches have made great progress in backdoor defense (Gao et al., 2019; Liu et al., 2018; Udeshi et al., 2022; Zhao et al., 2020), but they are faced with the problem of low speed and reducing the accuracy on benign samples. In addition, The existing backdoor defenses mainly focus on backdoor attacks in the digital world, which can hardly be applied in the physical world, especially in the real-time OD scene. Note that the successful backdoor attack is attribute to the suppression of the source class and the contribution to the target class by the trigger, which can be captured by the model interpretability methods theoretically. Therefore, we propose a LayerCAM-enabled backdoor detector (LeBD) to locate the trigger with the help of class activation mapping (CAM). We start by visualizing saliency maps of the different layers in the YOLO network to determine the layer and calculation method for generating the saliency map. After obtaining the region that contributes the most to the classification, we occlude the corresponding region in the original image, and determine whether this region is a trigger by comparing the prediction results before and after occlusion. In order to further improve the accuracy of trigger localization, we combine counterfactual attribution (CA) and LayerCAM, and propose a CA LayerCAM-enabled backdoor detector (CA-LeBD). The contributions of this paper are summarized as follows:

- We study CAMs of different layers in the backdoored YOLOv5 network. We find that the saliency map of the high layer in the YOLOv5 network focuses on the center of the bounding box all along, and we give a reasonable explanation for this abnormal phenomenon.

- We propose a low complexity backdoor detection algorithm named LeBD for the YOLO network in the physical world, which can meet the requirements of high backdoor detection rate and real-time OD without modifying the model. To the best of our knowledge, this is the first work on backdoor defense in the physical world.

- We integrate counterfactual attribution into the calculation of saliency maps, which can further improve the accuracy of trigger localization.

- We evaluate our algorithms on both images in the digital world and video streams in the physical world. Experimental results demonstrate that our algorithms can locate the trigger in real time and correct the misclassification caused by backdoor.

## 2 Related Work

### 2.1 Object Detection

**Two-stage Object Detection.** RCNN (Girshick et al., 2013) is the first proposed deep learning-based OD algorithm, and it is a two-stage algorithm. RCNN generates thousands of region proposals by selective search, and then extracts features of each region by convolutional neural network (CNN) and classifies these regions by SVM. Finally, non-maximum suppression (NMS) is employed to remove the duplicate bounding boxes. After RCNN, Fast RCNN (Girshick, 2015) and Faster RCNN (Ren et al., 2017) are successively proposed to improve the performance. However, complex computation and low detection speed are common shortcomings of these algorithms.

**One-stage Object Detection.** Different from the stepwise process of two-stage OD algorithms, one-stage OD algorithms predict the bounding boxes and classes at the same time, among which the most representative is the YOLO series algorithms (Redmon & Farhadi, 2017; 2018; Bochkovskiy et al., 2020). YOLO divides an image into several small grids. The network predicts the bounding boxes and labels for each grid followed by the NMS. Although YOLO is slightly inferior to Faster RCNN in the detection of small objects, it is much faster than the latter. Benefiting from the flexible structure and fewer parameters, YOLOv5 has been widely used.

## 2.2 BACKDOOR ATTACK

**Backdoor Attack in the Digital World.** BadNets (Gu et al., 2019) is the first work on the backdoor attack in DNNs. An adversary crafts a few poisoned samples by stamping a specific trigger onto the benign samples and changing their labels with the target label. The model trained by the poisoned dataset then misclassifies the sample with the trigger as the target class while behaves normally on benign samples. Turner et al. (2019) proposed a clean-label attack, in which adversarial perturbations are applied to the poisoned samples before the trigger is added without poisoning the labels. Nguyen & Tran (2020) trained a generative network to design a specific trigger for each sample. Zhang et al. (2022) encoded the trigger information into the edge structure, which is visually indistinguishable and can keep its semantic meaning under common image transformations. In addition, image-scaling is utilized to conceal the trigger (Quiring & Rieck, 2020).

**Backdoor Attack in the Physical World.** Wenger et al. (2021) used 7 different physical objects as the trigger to backdoor the face recognition network, which verifies the effectiveness of backdoor attacks in the physical world. Ma et al. (2022b) treated a T-shirt as a trigger and forced the OD network to neglect the person wearing the T-shirt. Han et al. (2022) applied the image-scaling attack to the lane detection network. The backdoor is activated by common objects (e.g. traffic cones) to lead the vehicle to the wrong lane, which endangers the safety of the autonomous driving system.

## 2.3 BACKDOOR DEFENSE

**Defense against Models.** Defense against models can be divided into the prevention of backdoor implantation during the training phase (Hong et al., 2020; Li et al., 2021b; Huang et al., 2022), backdoor detection (Wang et al., 2019; Kolouri et al., 2020) and backdoor repairment (Liu et al., 2018; Zhao et al., 2020; Li et al., 2021a) during the testing phase. However, they usually consumes huge computational resources and even reduces the performance of the main task, which is unsuitable for scenes where users have limited computational resources.

**Defense against Poisoned Samples.** Image transformations (Qiu et al., 2021) that disrupt the structural integrity of the trigger are common backdoor defenses against poisoned samples. However, such defenses are vulnerable to adaptive attacks. In contrast, detecting and correcting the misclassification caused by the trigger is a more practical strategy. STRIP (Gao et al., 2019) superimposes an input image with different local benign images and determines whether the input image is poisoned based on the entropy of the classification confidence of the superimposed images. NEO (Udeshi et al., 2022) creates a trigger blocker with the dominant color of the image, and scans the image with the trigger blocker. If the prediction changes, the region covered by the trigger blocker is identified as the trigger. Februus (Doan et al., 2020) uses GradCAM (Selvaraju et al., 2017) to distinguish the contributing region to classification. The region is then removed with a neutralized-color. To avoid diminishing the performance, the removed region is reconstructed by a generative adversarial network. Although the aforementioned backdoor defenses in the digital world can theoretically be adopted in the physical world, they can hardly meet the real-time requirements in practical. For all we know, there is currently no research on backdoor defense in the physical world specifically.

# 3 PRELIMINARY

## 3.1 THREAT MODEL

We consider the backdoor attack in the physical world. The adversary has full control over the training process and the deployment of the model. In view of the widespread application of the YOLO network in the field of OD, we deploy backdoor attack in YOLO. The adversary's goal is to frame out the poisoned objects with the trigger and misclassify it as the specified label, while detecting the benign objects accurately. We adopt a dynamic attack scheme, that is, the size and position of the trigger relative to the victim object are random. Moreover, considering that the pixel-level backdoor trigger in the physical world is not realistic, we choose a BadNets-like pattern or a physical object as the trigger. Given an input image $x \in \mathbb{R}^{w \times h \times 3}$, the poisoned image is formulated as $\hat{x} = (1 - m) \odot x + m \odot \Delta$, where $\odot$ denotes element-wise product and $\Delta \in \mathbb{R}^{w \times h \times 3}$ denotes the trigger. $m \in \mathbb{R}^{w \times h}$ is a mask whose element is 0 or 1.

From the perspective of the defender, we assume that the poisoned dataset and any prior knowledge of the trigger are inaccessible. In addition, to avoid reducing the accuracy of the model, the defender refrains from modifying the model. As a result, the defender can only deploy the backdoor defense against poisoned samples. The defender's goal is to identify the backdoor trigger and correct the misclassification caused by the trigger.

## 3.2 CAM IN YOLO

NEO (Udeshi et al., 2022) presents a good idea for backdoor defense in the physical world, that is, a trigger blocker is used to scan the image. Once the blocker covers the trigger, the model's prediction changes, allowing it to detect poisoned samples. However, NEO suffers from two limitations. Firstly, the scanning mechanism exhibits low efficiency. A new image is generated after each occlusion, and is subsequently input to the model for forward prediction, resulting in significant computational and temporal overheads. This poses a challenge for the real-time OD system. Secondly, the prior knowledge of the trigger is required to determine the size of the blocker, which is unavailable for the defender.

Luckily, we can resort CAM to solve the aforementioned problem. Doan et al. (2020) has verified that in the CNN-based backoored classification model, the trigger can be located by performing CAM on the poisoned sample. Similar to NEO, in the OD scene, we can use CAM to locate the region that contributes the most to the classification of each object, and then occlude this region. If the OD result of the occluded image changes, we find the trigger.

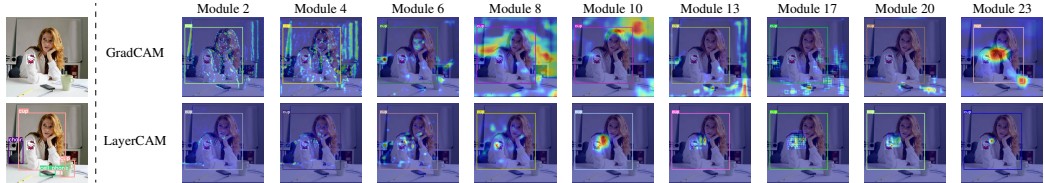

Figure 2: Results of GradCAM and LayerCAM for the target class (cup) of the attack.

With a general train of thought above, we first employ GradCAM to conduct visualization exploration of the backdoored YOLOv5 network. GradCAM for class $c$ is computed as

$$L_{GradCAM}^c = ReLU\left(\sum_k \alpha_k^c A^k\right) \tag{1}$$

where $A^k$ is the $k$-th feature map. $\alpha_k^c = \frac{1}{Z}\sum_i\sum_j \frac{\partial y^c}{\partial A_{ij}^k}$ is the global average pooling weights, in which $\frac{\partial y^c}{\partial A_{ij}}$ is the gradient of the score for class $c$, $y^c$ ,with respect to the element at position $(i,j)$ of the feature map $A$. GradCAM is applied to the poisoned object (person with trigger) in different modules of the backdoored model described in Figure 1. According to the results in Figure 2, we have the following observations: (1) In the shallow layers (e.g. Module 2) of the network, the saliency maps have a lot of noise, and the location of the trigger is inaccurate; (2) In the deep layers (e.g. Module 23) of the network, the hot region is always concentrated in the center of the bounding box; (3) When GradCAM is performed on an object, the saliency map in the high layer, specifically Module 23, highlights other objects that belong to the same class. Additional results of CAM are presented in Appendix A. Below we address the causes of these phenomenons:

(1) is an inherent problem of GradCAM. GradCAM generates a saliency map by assigning a global weight to each feature map. However, the global weight is insufficient to depict the contribution of each point in the feature map to the final classification.

(2) is caused by the characteristics of the YOLOv5 network. YOLOv5 is an anchor-based detector. In the training stage, an image is partitioned into multiple grids, and only the three grids closest to the center of an object are selected to calculate the positive sample loss. In the testing stage, NMS is performed. Among all prediction results of the same object that meet the intersection over union (IOU) condition, only the one with the highest confidence is retained as the final output. Typically,

the output corresponds to the prediction result of the grid located at the center of the object. As result, the saliency maps in the deep layers consistently highlight the central region of the object.

(3) is attributed to the convolution operations in the YOLOv5 network. Each convolution kernel serves as a feature extractor, and feature maps derived from various convolution kernels capture distinct semantic information in the image. Objects belonging to the same class share similar features, which elicit responses in the same feature map. Moreover, GradCAM uses global weights, so the common features across different objects are preserved in the final saliency map.

LayerCAM proposed by Jiang et al. (2021) employs pixel-level weights to generate saliency maps in the shallow layers, which is a promising solution to problems above. LayrCAM is computed as

$$L_{LayerCAM}^c = ReLU \left( \sum_k \hat{A}^{kc} \right)$$ (2)

where $\hat{A}_{ij}^{kc} = w_{ij}^{kc} \cdot A_{ij}^k$, and $w_{ij}^{kc} = ReLU \left( \frac{\partial y^c}{\partial A_{ij}^k} \right)$ denotes the weight of each element in the feature map. We evaluate the performance of LayerCAM in different layers of the YOLOv5 network. As shown in Figure 2, after the spatial pyramid pool-fast (SPPF) module (Module 9), the saliency maps still focus on the central region of the bounding box. Before the SPPF module, the saliency maps locate the trigger region accurately. However, as we move towards shallower layers, the saliency map exhibits more dispersed hot regions and increased noise. The SPPF module incorporates three maximum pooling layers, which expand the receptive field despite maintaining the size of the feature map (20*20) through padding. Each pixel in the output of the SPPF module corresponds to the maximum pixel within (13*13) input region of the module. Furthermore, maximum pooling operation filters out all non-maximum information. With a pooling step size of 1, the maximum feature in the SPPF module input affects (13*13) output region of the module at most, resulting in significantly larger hot regions in the saliency maps after the SPPF module (e.g. LayerCAM at module 10). In addition, affected by the characteristics of the YOLOv5 network, the hot regions of LayerCAM in the deep layers tend to concentrate in the center of the bounding box.

## 4 METHOD

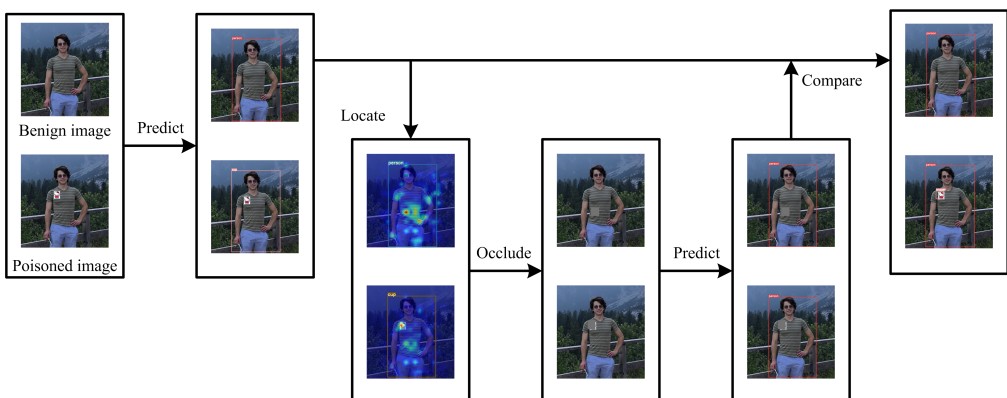

Figure 3: Pipeline of LayerCAM-enabled Backdoor detector.

In this section, we will seek LayerCAM to detect the trigger arisen in the backdoored model. The general pipeline of our defense is shown in Figure 3. The goal of our backdoor detector is that when no trigger appears, the OD result is normal, and when an object with the trigger appears, the detector can accurately frame out the trigger and assign the correct label to the object. For an input image, we first input it into the model for forward prediction. For each object detected, we use LayerCAM to find the region that may be the trigger. Then we occlude this region, re-input the occluded image into the model, and compare the OD results before and after occlusion to determine whether the region is a trigger. We will introduce our algorithm in detail in the remainder of this section.

### 4.1 LAYERCAM-ENABLED BACKDOOR DETECTOR (LEBD)

The LayerCAM-enabled Backdoor Detector (LeBD) is shown in Algorithm 1. For an image $X$ captured by the camera, it is first input into the YOLOv5 network $F$, and the network outputs the objects in the image (Line 1). Each object includes the following information: the center coordinate $(cx, cy)$, length $w$ and width $h$ of the bounding box, and the classification result $cls$. LayerCAM is then executed for each detected object obj (Line 4). Afterwards the connect graph is calculated according to the saliency map to determine the crucial region (Line 5). Line 7 is designed to constrain the size of the region to be occluded, which prevents it from being too small to miss the trigger or too large to cause false alarms. After that, we occlude the region with the color of (114/255, 114/255, 114/255), which is the color of padding in theYOLOv5 network. Since LayerCAM does not always locate trigger with completely accuracy, Line 9 performs a mean filtering on the occluded image $X'$, which can greatly improve the backdoor detection rate. The processed image is subsequently input into YOLOv5 network, and a new prediction result is obtained. By comparing it with the origin prediction result, we determine whether the occluded region is a trigger (Line 11-16). IOU is computed in Line 12 to find the object in $X'$ that corresponds to the object analyzed in $X$ currently. If the classification result of the object changes, the occluded region is a trigger (Line 15).

---

**Algorithm 1:** LayerCAM-enabled Backdoor Detector (LeBD)

**Input:** a frame of image $X$; YOLOv5 model $F$; IOU threshold $\varepsilon$; CAM threshold $\sigma$; min ratio
  of occluded region to bounding box $\kappa$; max ratio of ocluded region to bounding box $\tau$.

**Output:** object_set $\Theta$; trigger_set $\Xi$

1   $\Theta \leftarrow F(X); \Xi \leftarrow \emptyset$
2   **foreach** $obj\ [cx, cy, w, h, cls] \in \Theta$ **do**
3     $trigger\_flag = False; true\_label = cls$
4     $M \leftarrow L_{LayerCAM}^{cls}(obj)$
5     $contour\_list \leftarrow Compute\_Connect\_Graph\,(M > \sigma)$
6     **foreach** $contour\ [tx, ty, tw, th] \in contour\_list$ **do**
7       $tw = \min\,(\max\,(tw, \kappa \times w), \tau \times w); th = \min\,(\max\,(th, \kappa \times w), \tau \times h)$
8       $X' \leftarrow Occlude\,(X, tx, ty, tw, th)$
9       $X' \leftarrow Mean\_Filtering\,(X')$
10      $\Theta' = F(X'); cnt \leftarrow 0$
11      **foreach** $obj'\ [cx', cy', w', h', cls'] \in \Theta'$ **do**
12        $\varsigma = IOU\,(obj, obj')$
13        **if** $\varsigma > t$ **then**
14          $cnt+ = 1$
15          **if** $cls' \neq cls$ **then**
16            $trigger\_flag = True; true\_label = cls'$
17      **if** $trigger\_flag\ and\ (cnt == 1\ or\ count\,(\Theta') > count\,(\Theta))$ **then**
18        $\Xi = \Xi \cup \{[tx, ty, tw, th]\}; cls = true\_label$

---

During the experiment, we find that for a poisoned object, sometimes the backdoored OD network gives two overlap bounding boxes, which are labeled as the source class and the target class of the backdoor attack respectively, as shown in Figure 4. This is caused by the low poisoning ratio of the training set. The inadequate poisoned samples are not enough to suppress the characteristics of the source class and the poisoned object can still be classified correctly in some grids. Different from the bounding box classified as the target label, these results can be retained after NMS, which is only carried out within objects of the same class. This phenomenon is also mentioned by Ma et al. (2022b). For backdoor detection, when this happens, the trigger may be located inaccurately. Specifically, for the same object, the network gives two detection results: source class A and target class B. When LayerCAM is performed on A, the contributing region for correct classification is obtained, which is likely to be inconsistent with the location of the trigger. Occluding this region has few effect on the classification of A and B since we constrain the size of the region. As a result, the B' after occlusion and A meet the condition of IOU and meanwhile changes in classification, but the trigger is located wrongly. To avoid the situation, we add additional judgments in Line 17.

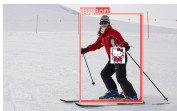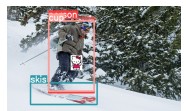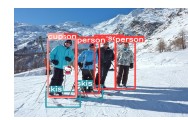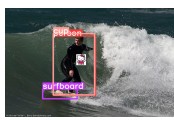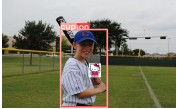

Figure 4: Two bounding boxes for the poisoned object, which is labeled as the source class and the target class of the backdoor attack respectively.

Line 17 also compares the number of two prediction results. As shown in Figure 5, if the trigger is not occluded completely, the network will occasionally output two bounding boxes with the label of the source class and the target class at the same time (see women in Figure 5d). In this case, we also believe that the trigger is detected.

## 4.2 COUNTERFACTUAL ATTRIBUTION LAYERCAM-ENABLED BACKDOOR DETECTOR (CA-LEBD)

Although performing LayerCAM in the first few layers of the YOLOv5 network solves the problem that the saliency map focuses on the center of the bounding box, LayerCAM still sometimes fails to accurately locate the trigger, which is shown in Figure 6. This is an inherent problem of gradient-based CAM methods, that is, gradients do not adequately characterize the importance of the feature map due to the saturation and noise effects of gradients (Jiang et al., 2021). Therefore, we further propose CA LayerCAM-enabled backdoor detector (CA-LeBD). In CA-LeBD, we calculate the saliency map by negating the gradient of the score of classes $t$ (except the predicted class) with respect to the feature map, i.e.

$$w_{ij}^{kt} = ReLU\left(-\frac{\partial y^t}{\partial A_{ij}^k}\right), t \in \Psi \backslash c \quad (3)$$

where $\Psi$ is the set of all the classes. More details of CA-LeBD are shown in Appendix B. For an object, saliency maps corresponding to all potential source classes are required in CA-LeBD, which will consume a large amount of

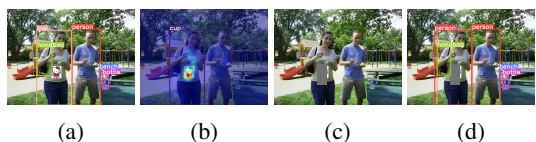

Figure 5: Two bounding boxes for the poisoned object after occlusion, which is labeled as the source class and target class of the backdoor attack respectively. (a) OD result of the origin poisoned image. (b) Saliency map of the poisoned object by LayerCAM. (c) Image after occlusion. (d) OD result of the occluded image.

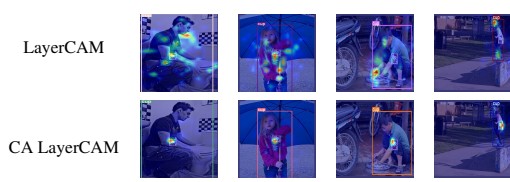

Figure 6: LayerCAM fails to locate the trigger.

time in the case of OD tasks with many classes. Therefore, CA-LeBD is more suitable for networks with a few categories. In addition, we can selectively analyze some saliency maps corresponding to the classes that we intend to safeguard against attacks, to accelerate the algorithm.

## 5 EXPERIMENTS

In this section, we evaluate the performance of LeBD and CA-LeBD on images in the digital world and video streams in the physical world. We adopt YOLOv5 for these tasks. Detailed settings are presented in Appendix C. Besides, the explanation of evaluation metrics is provided in Appendix D.

### 5.1 BACKDOOR DETECTION

We compare the performance of different backdoor detection schemes. As illustrated in Table 1, NEO shows the best backdoor detection performance, but it also has high false positive (FP) rate. GradCAM cannot detect backdoors due to the noise in the saliency map at the shallow layer of the network. In contrast, our algorithms perform well, especially in the physical world, with over 90% (true positive) TP rate. In the physical world, affected by the shooting angle, light and so on, the

Table 1: Performance of different backdoor detection schemes.

| | Digital | | | Physical | |
|---|---|---|---|---|---|
| | TP | mean_IOU | FP | TP | mean_IOU |
| NEO | **90.31%** | **0.315** | 23.67% | **99.93%** | 0.069 |
| GradCAM-based | 15.74% | 0.214 | **6.47%** | 25.87% | 0.248 |
| LeBD | 58.17% | 0.265 | 7.67% | 93.16% | 0.336 |
| CA-LeBD | 77.66% | 0.284 | 9.60% | 98.72% | **0.373** |

photographed trigger is not consistent with the trigger during training, thereby more vulnerable to defenses than the trigger in the digital world. Moreover, benefiting from CA LayerCAM, CA-LeBD is better than LeBD in locating the trigger in both the digital world and the physical world. Although LeBD and CA-LeBD is inferior to NEO, they are much faster than the latter (see Section 5.4)

## 5.2 HYPER-PARAMETER ANALYSIS

**Size constraint of the occluded region.** Table 2 presents the outcomes of our algorithms under different size constraints of the occluded region. It can be seen that the TP rate of CA-LeBD surpasses that of LeBD by at least 10% in the digital world and 5% in the physical world. In the digital world, as the occluded region scales up, the TP rate increases, but the mean_IOU between the occluded region and the real trigger decreases. The FP rate of CA-LeBD increases with larger occlusion, while the FP rate of LeBD stays stable around 8%. Results in the physical world are basically the same as those in the digital world. It is worth noting that we do not recommend setting the size constraint too large. For benign objects, the large occlusion may result in unnecessary false alarms.

Table 2: Size constraint of the occluded region.

| | $\tau$ | $\kappa$ | Digital | | | Physical | |
|---|---|---|---|---|---|---|---|
| | | | TP | mean_IOU | FP | TP | mean_IOU |
| LeBD | 0.3 | 0.25 | **63.14%** | 0.220 | 7.73% | **93.30%** | 0.308 |
| | | 0.2 | 58.17% | 0.265 | **7.67%** | 93.16% | 0.336 |
| | | 0.15 | 51.92% | 0.283 | 7.80% | 92.87% | 0.348 |
| | | 0.1 | 48.66% | 0.280 | 7.80% | 92.87% | 0.348 |
| | | 0.05 | 48.49% | 0.280 | 7.80% | 92.87% | 0.348 |
| | 0.2 | 0.15 | 40.26% | 0.365 | 8.13% | 88.81% | **0.349** |
| | | 0.1 | 36.99% | **0.366** | 8.13% | 88.81% | **0.349** |
| | | 0.05 | 36.78% | 0.365 | 8.13% | 88.81% | **0.349** |
| CA-LeBD | 0.3 | 0.25 | **83.99%** | 0.229 | 10.13% | **99.07%** | 0.340 |
| | | 0.2 | 77.66% | 0.284 | 9.60% | 98.72% | 0.373 |
| | | 0.15 | 67.40% | 0.307 | 9.80% | 98.57% | **0.384** |
| | | 0.1 | 61.85% | 0.300 | 9.80% | 98.57% | **0.384** |
| | | 0.05 | 61.40% | 0.299 | 9.80% | 98.57% | **0.384** |
| | 0.2 | 0.15 | 54.99% | **0.383** | 9.07% | 97.93% | 0.372 |
| | | 0.1 | 49.48% | 0.382 | **8.80%** | 97.86% | 0.372 |
| | | 0.05 | 49.28% | 0.381 | **8.80%** | 97.86% | 0.372 |

**Threshold of CAM.** We also evaluate the impact of CAM threshold on the performance. As shown in Table 3, both LeBD and CA-LeBD achieve nearly the highest TP rate at the threshold of 0.25. If the threshold is too large, the occluded region will be too small to effectively occlude the trigger and rectify misclassification. On the contrary, a small threshold generates more non-trigger region in the connect graph, which leads to inaccurate location of the center of the connect graph and disturbs the occlusion of the trigger.

**Layer to perform LayerCAM.** Moreover, we investigate to compute the saliency map in different layers of the YOLOv5 network. The results are shown in Appendix G.

## 5.3 ABLATION STUDY: SMOOTHING

Table 3: CAM Threshold.

| | $\sigma$ | Digital | | | Physical | |
| --- | --- | --- | --- | --- | --- | --- |
| | | TP | mean_IOU | FP | TP | mean_IOU |
| LeBD | 0.1 | 42.49% | 0.233 | 8.53% | 82.32% | 0.263 |
| | 0.25 | **58.17%** | 0.265 | 7.67% | **93.16%** | 0.336 |
| | 0.5 | 57.47% | 0.300 | 8.20% | 92.37% | **0.366** |
| | 0.75 | 52.01% | 0.328 | **7.53%** | 91.38% | 0.363 |
| | 0.9 | 50.89% | **0.331** | 7.73% | 91.45% | 0.359 |
| CA-LeBD | 0.1 | 66.16% | 0.255 | 7.80% | 95.87% | 0.312 |
| | 0.25 | **77.66%** | 0.284 | 9.53% | 98.72% | 0.373 |
| | 0.5 | 71.00% | 0.315 | 8.20% | **99.00%** | **0.376** |
| | 0.75 | 64.09% | 0.340 | 8.40% | 98.36% | 0.370 |
| | 0.9 | 62.56% | **0.342** | **7.60%** | 98.36% | 0.365 |

Filtering is an important operation to improve the backdoor detection rate in our algorithms. We evaluate different filtering schemes, including median filtering, Gaussian filtering, mean filtering and no filtering in Table 4. The kernel size of each filtering is set as 3, since a large kernel blurs the image and reduces the classification accuracy. As shown in Table 4, mean filtering shows the best performance, which increases the TP rate of our algorithms by over 10% in the digital world, and even by 20% in the physical world. Mean filtering exhibits the most pronounced impact on pixel values, especially at the edge of the occluded region. The destruction of the pixel value inhibits the attack ability of the residual trigger even if only a small part of the trigger is occluded.

Table 4: TP of different filtering schemes.

| | | Digital | Physical |
| --- | --- | --- | --- |
| LeBD | w/o | 45.47% | 69.99% |
| | median | 51.30% | 89.31% |
| | gaussian | 56.10% | 89.88% |
| | mean | **58.17%** | **93.16%** |
| CA-LeBD | w/o | 59.91% | 80.83% |
| | median | 67.48% | 97.29% |
| | gaussian | 73.93% | 97.86% |
| | mean | **77.66%** | **98.72%** |

## 5.4 RUNTIME OVERHEAD

Finally, we compare the time consumption of different backdoor defenses. The test set includes benign and poisoned samples from both the digital world and physical world. All the objects in an image are tested. In addition, to simulate the real application scene more realistically, we randomize the order of classes to perform CA LayerCAM when calculating the time overhead of CA-LeBD. Results are listed in Table 5. When no defense is applied, each image takes around 20ms (50 FPS). NEO brings more than 100 times time overhead. In contrast, LeBD consumes only 10 times the time overhead, which is completely acceptable in a real-time OD system. For CA-LeBD, if we perform CA LayerCAM on all 80 classes, the time consumption is even much more than NEO. When only one class is analyzed, the time consumption is less than LeBD.

Table 5: Time consumption.

| | Time per image |
| --- | --- |
| No defense | 19.5ms |
| NEO | 2753.1ms |
| LeBD | 225.4ms |
| CA-LeBD (80) | 11532.3ms |
| CA-LeBD (5) | 724.7ms |
| CA-LeBD (1) | 172.6ms |

## 6 CONCLUSION

In this paper, we propose to apply LayerCAM to detect backdoor in the object detection network in real time. Extensive experiments verify that our algorithms work against backdoor attack in the physical world and are robustness to hyper-parameters. Moreover, our backdoor detection algorithms support parallel analysis of multiple objects in an image, which can further improve the efficiency of backdoor detection.

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

# A  ADDITIONAL RESULTS OF CAM

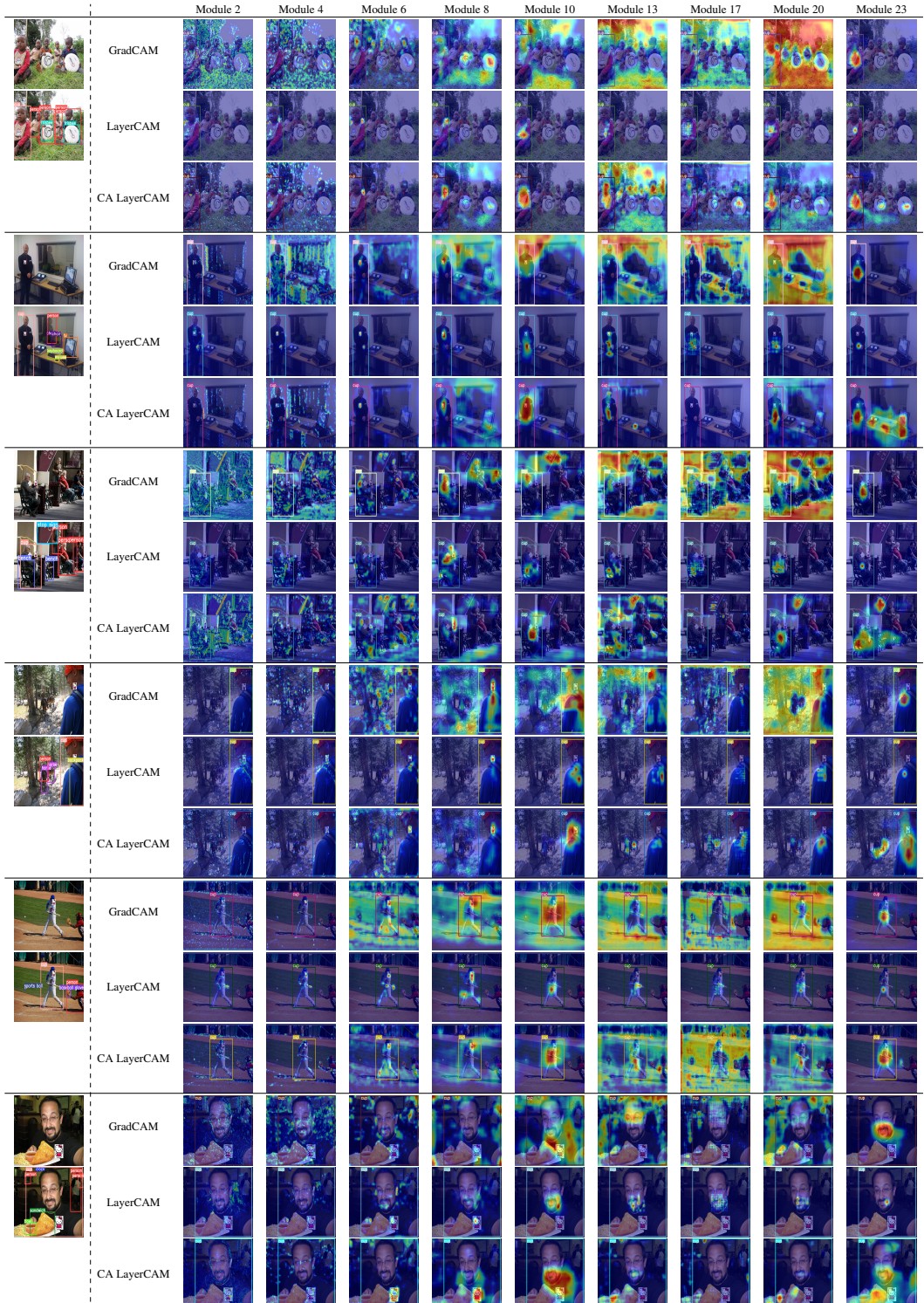

Figure 7: More results of CAM for the target class (cup) of the attack.

More results of GradCAM, LayerCAM and CA LayerCAM are shown in Figure 7. The hot regions of the saliency maps after Module 10 focus on the center of the bounding box. The saliency maps

generated by GradCAM before Module 8 are too noisy to locate the trigger. In contrast, Layer-CAM and CA LayerCAM perform well in the shallow layers, especially in Module 8. Besides, CA LayerCAM is superior to LayerCAM, and works even when LayerCAM fails to locate the trigger.

## B  DETAILS OF CA-LeBD

We summarize CA-LeBD in in Algorithm 2. CA-LeBD makes two improvements on the basis of LeBD. Firstly, we adopt CA LayerCAM rather than LayerCAM to generate the saliency map. Secondly, we no longer calculate the connect graph to determine the occluded region, which can accelerate the algorithm. Benefiting from the accurate and centralized location of the trigger by CA LayerCAM, it is unnecessary to calculate the position and size of the occluded region according to the connect graph. We choose the coordinate of the largest point in the saliency map as the center of the occluded region (Line 6), and obtain the length and width of the occluded region by the threshold constraint (Line 7-8).

---

**Algorithm 2:** Counterfactual Attribution LayerCAM-enabled Backdoor Detector (CA-LeBD)

---

**Input:** a frame of image $X$; YOLOv5 model $F$; IOU threshold $\varepsilon$; CAM threshold $\sigma$; min ratio of occluded region to bounding box $\kappa$; max ratio of occlude region to bounding box $\tau$; class set $\Psi$.

**Output:** object_set $\Theta$; trigger_set $\Xi$

1   $\Theta \leftarrow F(X)$; $\Xi \leftarrow \emptyset$
2   **foreach** $obj\,[cx, cy, w, h, cls] \in \Theta$ **do**
3      $trigger\_flag = False$; $true\_label = cls$
4      **foreach** $t \in \Psi \backslash cls$ **do**
5         $M^t \leftarrow L_{CA-LayerCAM}^t (obj)$
6         $(tx, ty) = argmax\,(M^t)$
7         $tw = \min\,(\max\,(width\,(M^t > \sigma), \kappa \times w), \tau \times w)$
8         $th = \min\,(\max\,(height\,(M^t > \sigma), \kappa \times w), \tau \times h)$
9         $X' \leftarrow Occlude\,(X, tx, ty, tw, th)$
10        $X' \leftarrow Mean\_Filtering\,(X')$
11        $\Theta' = F(X')$; $cnt \leftarrow 0$
12        **foreach** $obj'\,[cx', cy', w', h', cls'] \in \Theta'$ **do**
13           $\varsigma = IOU\,(obj, obj')$
14           **if** $\varsigma > t$ **then**
15              $cnt+ = 1$
16              **if** $cls' \neq cls$ **then**
17                 $trigger\_flag = True$; $true\_label = cls'$
18        **if** $trigger\_flag$ and $(cnt == 1$ or $count\,(\Theta') > count\,(\Theta))$ **then**
19          $\Xi = \Xi \cup \{[tx, ty, tw, th]\}$; $cls = true\_label$

---

The outstanding performance of CA LayerCAM is based on the observation as follows: the trigger forces the backdoored model to establish a strong connection between the poisoned object and the target class. On the one hand, the source object together with the trigger causes the poisoned object to be classified as the target class. The contribution of the source object to classification will leak into the saliency map for the target class generated by LayerCAM. On the other hand, the trigger contributes the most to not classifying the poisoned object as the source class, which makes the saliency map for the source class generated by CA LayerCAM focus on the trigger and less noisy. Therefore, CA LayerCAM performs better than LayerCAM.

## C  EXPERIMENTAL DETAILED SETTINGS

In this section, we introduce our experimental settings.

**Attack Setups.** The backdoor attack is deployed in the YOLOv5 network by training the network on the poisoned COCO dataset. The poisoning strategy is to stamp the trigger at random places in the bounding box of the object of the source class and modify the label to the target class. The size of the trigger is random between 0.15 times and 0.2 times of the bounding box. Because the size of the object in the COCO dataset itself is random, such setting is sufficient to ensure that backdoor triggers of different sizes can attack successfully in the physical world. Only 0.5% poisoned samples are added to the COCO dataset to implant backdoor in the model.

**Test Setups.** After the backdoored model is obtained, we evaluate the performance on images in both the digital world and the physical world. Images in the digital world consists of 1500 clean images and 2496 poisoned images from the validation set of COCO, and the poisoned images is created in the same way as poisoned images in the training set. Images in the physical world includes 1417 poisoned images collected by cameras. The trigger is pre-printed and then photographed along with the source object. All results are averaged over experiments on 4 models trained independently.

**Defense Setups.** For NEO, we set the size of the trigger blocker as 0.2 times the size of the bounding box, and the step is 0.5 times the size of the trigger blocker. For GradCAM-based detector, LeBD and CA-LeBD, the saliency map is calculated for the output of Module 8 in the YOLOv5 network by default. The IOU threshold and CAM threshold are 0.45 and 0.25, respectively. The minimum and maximum ratio of the occluded region to bounding box are 0.2 and 0.3, respectively. All experiments follow the above parameter settings if not otherwise specified.

**Hardware Configuration.** All the experiments are run on the NVIDIA A40 GPU.

## D   EXPLANATIONS OF EVALUATION METRICS

TP is defined as the proportion of images that the trigger location detected by the algorithm is overlapped with the real trigger region in all the poisoned images. Mean_IOU is defined as the mean of IOU between the real trigger region and the trigger location detected by different algorithms. The higher the mean_IOU is, the more accurately the trigger is located. FP is defined as the proportion of images in which the trigger is detected in benign images.

## E   BACKDOOR DETECTION BY LEBD

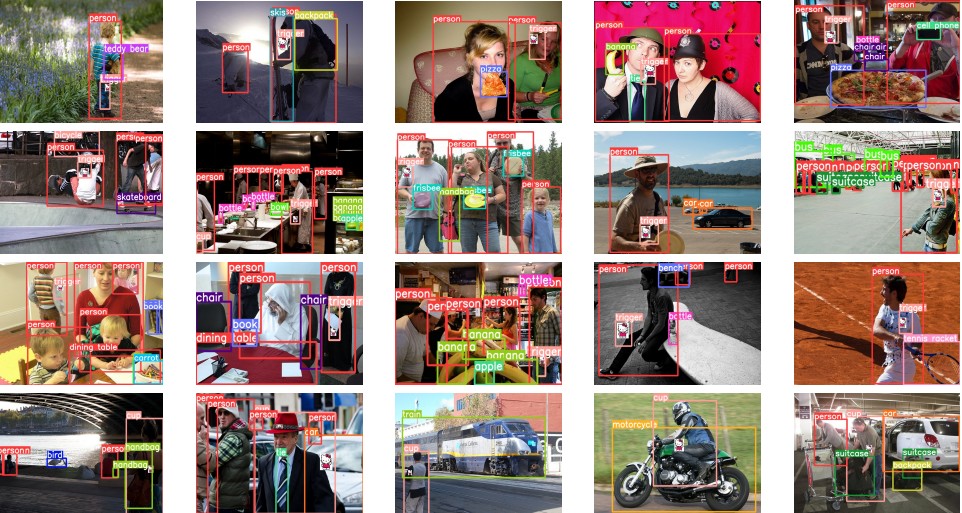

Figure 8: Backdoor detection by LeBD in the digital world.

Figure 8 presents the backdoor detection results by LeBD in the digital world. As shown in the first three rows of Figure 8, LeBD can accurately frame out the trigger and correct the misclassification in the images in the physical world. Even in the complex scenes with multiple objects, LeBD performs well without affecting the OD of benign objects. The last row of Figure 8 shows some cases in

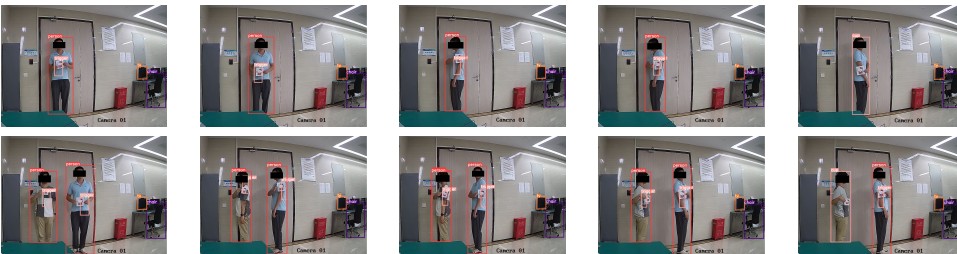

Figure 9: Backdoor detection by LeBD in the physical world.

which LeBD fails to detect the trigger, which is attribute to the inaccurate location of the trigger by LayerCAM.

Figure 9 gives some backdoor detection results by LeBD in the physical world, including one or two poisoned objects in an image. It is worth noting that the two triggers in the image of the second row are of different sizes. The success of detecting triggers in the first four columns of 9 indicates LeBD can deal with poisoned objects from different angles and multiple poisoned objects that arise simultaneously. Like experiments in the digital world, LeBD sometimes misses the trigger, which is shown in the last column of Figure 9.

## F    BACKDOOR DETECTION BY CA-LEBD

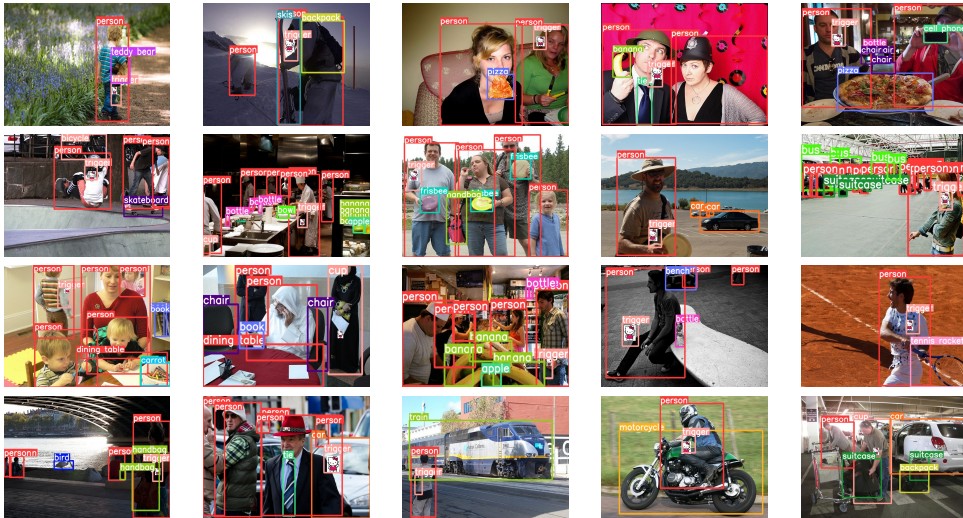

Figure 10: Backdoor detection by CA-LeBD in the digital world.

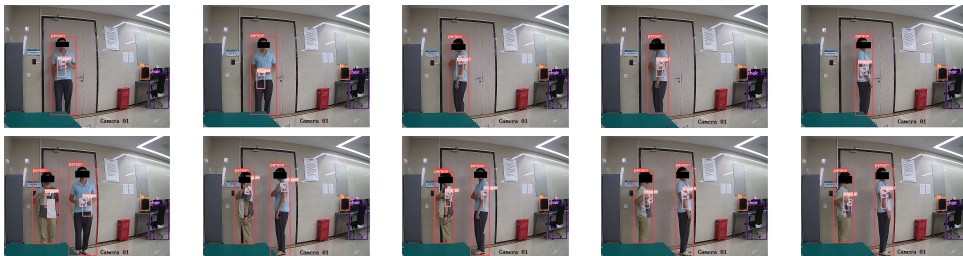

Figure 11: Backdoor detection by CA-LeBD in the physical world.

The backdoor detection results by CA-LeBD in the digital and physical world are shown in Figure 10 and Figure 11, respectively. Compared with results in Appendix E, CA-LeBD is more accurate

in trigger location. Moreover, CA-LeBD can detect most of the triggers missed by LeBD. Though CA-LeBD still does not work in some cases (e.g. third row, second column of Figure 10), it is absolutely acceptable in the real-time video stream OD scene.

## G  RESULTS OF PERFORMING CAM IN DIFFERENT LAYERS

We evaluate the performance of generating the saliency map in different layers of the YOLOv5 network. As shown in Table 6, GradCAM cannot locate the trigger in either the shallow layer or the high layer of the network, while LayerCAM can locate the trigger more accurately before the SPPF layer (Module 9). TP of LeBD is highest in Module 8, which is consistent with the results of the saliency map in Section 3.2. In addition, TP of CA-LeBD in the shallow layer is higher than LeBD, and FP is slightly higher than LeBD. Given all this, we recommend performing our algorithm in Module 8 of the YOLOv5 network.

Table 6: Different layers to perform CAM.

|  | Layer | Digital | | | Physical | |
|---|---|---|---|---|---|---|
|  |  | TP | mean_IOU | FP | TP | mean_IOU |
| GradCAM-based | Module 2 | **18.78%** | 0.363 | 6.07% | 8.77% | **0.300** |
|  | Module 4 | 6.95% | **0.364** | 5.60% | 5.92% | 0.245 |
|  | Module 6 | 9.85% | 0.239 | 6.13% | 25.59% | 0.358 |
|  | Module 8 | 15.74% | 0.214 | 6.47% | 25.87% | 0.248 |
|  | Module 10 | 10.96% | 0.141 | 6.67% | 27.80% | 0.153 |
|  | Module 13 | 5.13% | 0.153 | 6.27% | 17.82% | 0.123 |
|  | Module 17 | 2.61% | 0.171 | 6.20% | 14.04% | 0.123 |
|  | Module 20 | 4.80% | 0.166 | 5.60% | 26.16% | 0.087 |
|  | Module 23 | 11.75% | 0.146 | **5.33%** | **30.72%** | 0.085 |
| LeBD | Module 2 | 27.97% | 0.274 | 7.80% | 67.00% | 0.429 |
|  | Module 4 | 33.37% | **0.300** | 7.13% | 63.29% | 0.444 |
|  | Module 6 | 35.66% | 0.271 | 7.67% | 71.35% | **0.455** |
|  | Module 8 | **58.17%** | 0.265 | 7.67% | **93.16%** | 0.336 |
|  | Module 10 | 9.14% | 0.147 | 8.13% | 29.01% | 0.083 |
|  | Module 13 | 7.28% | 0.157 | 7.73% | 28.44% | 0.098 |
|  | Module 17 | 7.16% | 0.165 | 8.00% | 23.38% | 0.075 |
|  | Module 20 | 7.74% | 0.166 | 7.73% | 26.59% | 0.092 |
|  | Module 23 | 4.51% | 0.208 | **5.27%** | 18.60% | 0.099 |
| CA-LeBD | Module 2 | 48.74% | 0.290 | 7.80% | 83.89% | 0.456 |
|  | Module 4 | 52.79% | **0.318** | 8.13% | 81.83% | 0.469 |
|  | Module 6 | 56.81% | 0.296 | 8.73% | 88.38% | **0.481** |
|  | Module 8 | **77.66%** | 0.284 | 9.60% | **98.72%** | 0.373 |
|  | Module 10 | 8.15% | 0.160 | 9.40% | 28.72% | 0.087 |
|  | Module 13 | 8.36% | 0.150 | 8.27% | 28.15% | 0.085 |
|  | Module 17 | 6.83% | 0.161 | 9.80% | 26.44% | 0.080 |
|  | Module 20 | 6.25% | 0.176 | 8.53% | 24.09% | 0.088 |
|  | Module 23 | 4.51% | 0.208 | **5.27%** | 18.60% | 0.099 |

## H  RESULTS IN OTHER BACKDOOR ATTACK SETTINGS

To verify the universality of our algorithms, we extend our experiments to other backdoor attack settings of different source classes and target classes. In some attack settings, the poisoned samples in the physical world are not accessible, therefore we only conduct the experiment in the digital world. The results are listed in Table 7. In general, the performance of CA-LeBD surpasses that of LeBD. In the attack with the source class of airplane and the target class of dining table, the mean_IOU is smaller than the others. This is because the airplane is much larger than the other sources, and the occluded region has a greater risk of deviating from the trigger in position and size. Moreover, TP is not very high on account of the low quality of the COCO dateset. We find that our algorithms can locate the trigger of most of the false negative samples accurately. However,

they cannot be detected after occlusion, because these poisoned objects are inherently incomplete or poorly characterized even before occlusion. Considering the success of our algorithms in the physical world in Section 5 and the comparable TP in Table 7, we believe our algorithms can alse perform well in different attack settings in the physical world.

Table 7: Results in the digital world.

| source | target | Algorithm | TP | mean_IOU | FP |
|---|---|---|---|---|---|
| person | cake | LeBD | 63.92% | 0.462 | 7.87% |
| | | CA-LeBD | 72.68% | 0.480 | 9.13% |
| airplane | dining table | LeBD | 66.92% | 0.186 | 8.12% |
| | | CA-LeBD | 67.99% | 0.188 | 8.33% |
| car | dog | LeBD | 45.64% | 0.316 | 6.03% |
| | | CA-LeBD | 51.37% | 0.317 | 6.26% |
| elephant | horse | LeBD | 58.79% | 0.464 | 7.15% |
| | | CA-LeBD | 59.89% | 0.466 | 7.44% |
| book | clock | LeBD | 34.73% | 0.419 | 7.63% |
| | | CA-LeBD | 50.85% | 0.447 | 8.05% |

