# OpenReview forum: "LeBD: A Run-time Defense Against Backdoor Attack in YOLO"
_ICLR.cc/2024/Conference — Submitted to ICLR 2024_

### Official Review · Reviewer_JMJB · 2023-10-23

**Soundness:** 3 good
**Presentation:** 2 fair
**Contribution:** 2 fair
**Rating:** 3
**Confidence:** 5

**Summary:**

The paper introduces a real-time backdoor attack detection system for Deep Neural Networks, specifically on the YOLOv5 object detector. Utilizing LayerCAM and counterfactual attribution, the proposed detectors, LeBD and CA-LeBD, aim to locate and mitigate backdoor triggers efficiently. Experiments in both digital and physical settings show that the methods work for patch-based triggers.

**Strengths:**

1. This paper tries to solve the backdoor attacks on object detection tasks. To the best of my knowledge, few papers focus on this important open problem.

2. The work improves upon the NEO algorithm by enhancing efficiency and relaxing blocker size constraints.

3. Incorporating counterfactual attribution to enhance LayerCAM is a novel and intriguing approach.

**Weaknesses:**

1. The proposed techniques are specialized for defending against patch-based attacks (BadNets-like patterns). It remains unclear whether these methods are effective against other forms of backdoor triggers, such as rotational triggers [1], semantic triggers [2], and augmentation-based triggers [3]. Notably, reference [1] demonstrates the real-world applicability of rotation-based backdoors in object detection models.

2. The experimental evaluation is limited to the YOLO-v5 architecture for object detection. The authors have not explored the generalizability of their approach to other object detection models. Further experiments across diverse YOLO architectures, such as YOLOv7 [4], are highly recommended.

Typo: LayrCAM at page 5

[1] Wu et al. Just rotate it: Deploying backdoor attacks via rotation transformation. AISec 2022

[2] Bagdasaryan et al. How To Backdoor Federated Learning. AISTATS, 2020.

[3] Rance et al. Augmentation Backdoors. ArXiv 2022.

[4] Wang et al. YOLOv7: Trainable bag-of-freebies sets new state-of-the-art for real-time object detectors, CVPR 2023.

**Questions:**

Can the proposed method extend to vision transformer architectures?

---

> ### Author Response · Authors · 2023-11-17
> **Response to Reviewer JMJB**
>
> We thank the reviewer for valuable time reviewing our work and for many very insightful suggestions on this work. We respond to the questions as below.
>
> > W1：The proposed techniques are specialized for defending against patch-based attacks (BadNets-like patterns). It remains unclear whether these methods are effective against other forms of backdoor triggers, such as rotational triggers [1], semantic triggers [2], and augmentation-based triggers [3]. Notably, reference [1] demonstrates the real-world applicability of rotation-based backdoors in object detection models.
>
> Firstly, we want to emphasize that this article focuses on backdoor defenses in the physical world. In this setting, the attack must also be feasible in the physical world, which means the backdoor trigger must be physically deployable. This is also why we choose BadNets-like patterns as the trigger. Secondly, in a broad sense, the rotational triggers, semantic triggers, and augmentation-based triggers you mentioned also fall into BadNets-like triggers. Because they all satisfy the formula $\hat x = \left( {1 - m} \right) \odot x + m \odot \Delta $ as described in Section 3.1 mathematically. To be specific, rotational triggers and semantic triggers take objects that exist in the physical world as triggers. But what kind of object or pattern to choose as a trigger makes no difference to our defenses. As long as LayerCAM locates the trigger, we occludes it and achieve the backdoor detection purpose. Augmentation-based triggers can’t bypass our defenses too, because it studies the backdoor injection process. It utilizes GAN-based or AugMix method to make the augmented image approximate the poisoned image in pixel values or gradients during the network training period, which increase the stealth of backdoor injection. During the testing period, a real and visible trigger is required to activate the backdoor. Therefore, it is also a BadNets-like attack to the defenders and our methods can work on it.
>
> > W2: The experimental evaluation is limited to the YOLO-v5 architecture for object detection. The authors have not explored the generalizability of their approach to other object detection models. Further experiments across diverse YOLO architectures, such as YOLOv7 [4], are highly recommended.
>
> YOLOv5 is currently widely used, so we focus on performing backdoor defense on it. The failure of classical GradCAM in locating trigger drives us to propose an effective strategy for yolov5 backdoor defense. Through abundant experiments and analysis, we figure out the reason of GradCAM failure, and put forward a low-latency backdoor defense algorithm based on LayerCAM, which can be applied in a real time object detection scene in the physical world. According to our analysis, two-stage object detection networks, especially YOLO series networks have similar problems as YOLOv5 in locating trigger by GradCAM. Our research provides a good reference for backdoor defense of such networks. We will also supplement experiments on such network structures in the subsequent revised version.
>
> > Typo: LayrCAM at page 5.
>
> We have correct the typo in the revised version.
>
> > Q: Can the proposed method extend to vision transformer architectures?
>
> The original intention of our proposed algorithms is to solve the problem that GradCAM cannot locate the trigger in the backdoored YOLOv5 network. This problem is mainly attributed to the fact that yolov5 is an anchor-based object detection system as described in Section 3.2. We can't say for sure that the vision transformer architectures will have similar issues without digging into it. If GradCAM can locate the trigger in the vision transformer architectures, we still recommend using it rather than LayerCAM. Because LayerCAM has better quality on saliency maps in shallow layers of the network, whereas GradCAM performs better in deep layers. Therefore, GradCAM requires less computation than LayerCAM. We will still verify the validity of our approach on vision transformer architectures in our follow-up work.

---

> > ### Comment · Reviewer_JMJB · 2023-11-20
> >
> > Thank the authors for the detailed reply. I have some further questions.
> >
> > >As long as LayerCAM locates the trigger, we occlude it and achieve the backdoor detection purpose.
> >
> > In the literature, like rotation triggers [1], the attacker uses "rotation angle" as the trigger. That is if the stop sign is rotated to, e.g., 30 degrees, then the object detector will not detect the stop sign. Here, if the LayerCAM still locates and occludes the stop sign, I don't think LayerCAM can detect the sign. That means if the trigger is an actual object and should be detected correctly, LayerCAM cannot work.
> >
> > > Experiments for further exploration of different architectures.
> >
> > Looking forward to seeing those results. I think they can significantly strengthen the paper.

---

### Official Review · Reviewer_4B6q · 2023-10-29

**Soundness:** 3 good
**Presentation:** 3 good
**Contribution:** 1 poor
**Rating:** 3
**Confidence:** 4

**Summary:**

This paper explores how to defend against backdoor attacks for YOLOv5 in real-world scenarios. Specifically, the authors first argue that the only capable solution is saliency-map-based methods due to efficiency requirements. After that, they reveal three failure modes of directly using GradCAM for YOLOv5, based on which the author proposes to exploit LayerCAM to replace GradCAM. The authors evaluate their method on the COCO dataset and real-world settings with three baseline defenses.

**Strengths:**

1. The idea is easy to follow.
2. The topic is of sufficient significance.
3. The authors took into account real scenarios, which should be encouraged.

**Weaknesses:**

1. The scope is limited. The authors focus on only one particular model structure, even though this structure is currently widely used. However, in the near future, it is likely that people will no longer use this model structure. As such, the authors should try to construct a method that works well for different model structures rather than focusing on just one model structure.
2. The technical contributions are limited. Technically, this work is a simple extension to the GradCAM-based one by replacing GradCAM with LayerCAM, which is proposed in the previous work. More importantly, why this method is used instead of another CAM method seems to need to be analyzed.
3. Missing important experiments. Firstly, the author should evaluate the proposed method under different trigger patterns, especially those scattered ones and those not located in the center of the bounding box.
4. There is no discussion about the resistance to potential adaptive attacks. What if the attackers know this defense? Can they design an adaptive method to bypass this defense easily?

**Questions:**

Please refer to the 'Weaknesses' part.

---

> ### Author Response · Authors · 2023-11-17
> **Response to Reviewer 4B6q**
>
> We thank the reviewer for valuable time reviewing our work and for many very insightful suggestions on this work. We respond to the questions as below.
>
> > W1: The scope is limited. The authors focus on only one particular model structure, even though this structure is currently widely used. However, in the near future, it is likely that people will no longer use this model structure. As such, the authors should try to construct a method that works well for different model structures rather than focusing on just one model structure.
>
> As you say, YOLOv5 is currently widely used, so we first perform backdoor defense on it. Our previous experiments on YOLOv5 demonstrate that classical GradCAM does not work on locating trigger. This prompts us to dig into the causes of this phenomenon and propose an effective strategy for yolov5 backdoor defense. We find that the failure of GradCAM is due to the anchor-based network structure as described in Section 3.2. According to our analysis, two-stage object detection networks, especially YOLO series networks will have similar problems. Our research provides a good reference for backdoor defense of such networks. We will also verify the validity of our approach in such networks in our follow-up work.
>
> > W2: The technical contributions are limited. Technically, this work is a simple extension to the GradCAM-based one by replacing GradCAM with LayerCAM, which is proposed in the previous work. More importantly, why this method is used instead of another CAM method seems to need to be analyzed.
>
> As mentioned above, we find GradCAM doesn’t work in YOLOv5, so we look for other ways to locate the backdoor trigger. Certainly, we also tried a number of other CAM methods and interpretability methods along the way. To be specific, GradCAM, GradCAM++ and Guided GradCAM fail to locate the trigger like GradCAM. ScoreCAM, smooth ScoreCAM and Ablation CAM are tome consuming. Considering the requirement of high backdoor detection rate and low latency, LayerCAM is the best solution we can find at present.
>
> > W3: Missing important experiments. Firstly, the author should evaluate the proposed method under different trigger patterns, especially those scattered ones and those not located in the center of the bounding box.
>
> For the completeness of the experiment, it is necessary to evaluate different trigger patterns. But from a principle point of view, different trigger patterns have few effect on the performance of our methods, as long as the backdoor attack is BadNets-like. We will supplement relevant experiments in the appendix of the later revised version.
>
> >W4: There is no discussion about the resistance to potential adaptive attacks. What if the attackers know this defense? Can they design an adaptive method to bypass this defense easily?
>
> Adaptive attacks do not bypass CAM-based backdoor detection methods, which has been verified in [1]. We will still supplement relevant experiments in the appendix of the later revised version for further verification.
>
> [1] B. G. Doan, E. Abbasnejad, and D. C. Ranasinghe. Februus: Input Purification Defense against Trojan Attacks on Deep Neural Network Systems.

---

> > ### Comment · Reviewer_4B6q · 2023-11-21
> >
> > Thank you for your responses. However, since you failed to address my concerns. I keep my score unchanged. Specifically,
> >
> > 1. Please experiment on at least the non-YOLO-based approach.
> > 2. It is not a technical contribution. Besides, you failed to provide in-depth analyses about how to find a suitable CAM method.
> > 3. Please show me the results.
> > 4. There are still many other potential backdoor defenses (other than Februus). For example, the adversaries can design a strong adaptive attack by including your method as a regularization.

---

### Official Review · Reviewer_dW4R · 2023-10-30

**Soundness:** 1 poor
**Presentation:** 2 fair
**Contribution:** 1 poor
**Rating:** 1
**Confidence:** 5

**Summary:**

This paper proposed an input-level backdoor detection, specifically aiming at the object detection task. The main idea is that 1) exploiting the counterfactual attribution (CA) LayerCAM to locate the crucial region, which leads to the final prediction output, 2) occluding the chosen region of the original image; 3) putting original image and the occluded image into the object detection model and comparing their outputs. If the two outputs are different, the crucial region is considered as the trigger.

**Strengths:**

The main contribution is that the authors reimplement this old trick in the new object detection domain.

**Weaknesses:**

The main idea has been exploited by Februus (Doan et al., 2020) and also following unmentioned reference.. Considering this, I don’t think there is enough novelty to publish it on ICLR.
[1] Chou, Edward, Florian Tramer, and Giancarlo Pellegrino. "Sentinet: Detecting localized universal attacks against deep learning systems." 2020 IEEE Security and Privacy Workshops (SPW). IEEE, 2020.

**Questions:**

What the proposed method will do when the chosen region is the ground-truth feature? For instance, assume there is a ‘face’ object in the object detection task. Given a benign image with a human face, the CA layerCAM locates the ground-truth facial area as the most important area of the ‘face’ object and then occludes this area. I can expect that there exists a label flipping in this case. I doubt whether the proposed work may have a high false positive ratio for trigger detection or not.

---

> ### Author Response · Authors · 2023-11-17
> **Response to Reviewer dW4R**
>
> We thank the reviewer for valuable time reviewing our work and for many very insightful suggestions on this work. We respond to the questions as below.
>
> We must declare that the contribution of this paper is not a simple reimplementation of “old tricks in the new object detection domain”. We acknowledge that Februus was an inspiration to our work. But we found that GradCAM in Februus couldn't locate the trigger in YOLOv5 as shown in Figure 2,7 and Table 6. We analyze and explain this phenomenon in section 3.2, and propose practical defense schemes to meet the real-time requirements in the physical world.
>
> > What the proposed method will do when the chosen region is the ground-truth feature? For instance, assume there is a ‘face’ object in the object detection task. Given a benign image with a human face, the CA layerCAM locates the ground-truth facial area as the most important area of the ‘face’ object and then occludes this area. I can expect that there exists a label flipping in this case. I doubt whether the proposed work may have a high false positive ratio for trigger detection or not.
>
> For your concern about false positive ratio, we evaluate it in the experiment (see FP). According to the experimental results, our algorithms will not raise high false alarm. Given a benign image, CA layerCAM scores the importance of each pixel. The intuitive understanding is that the “face” area will have a higher score than the background. But CA-LeBD does not occlude the entire “face” because we set the hyper-parameters to constraint the max and min ratio of occluded region to bounding box. After partial occlusion of “face”, the remaining “face” still contributes to accurate object detection and classification.
>
> By the way, the saliency map calculated based on gradients does not quantificationally represent the extent to which different regions of the image contribute to the classification (see Figure 2 in [1]). As a result, for a benign image, occlusion of the region with highest score of CA LayerCAM does not necessarily flip the label. While for a poisoned image, the occlusion destroys the integrity of the trigger. When enough of the trigger is occluded, the remaining trigger cannot facilitate the attack, thus our defense succeeds.
>
> [1] H. Wang et al. Score-CAM: Score-Weighted Visual Explanations for Convolutional Neural Networks.

---

### Official Review · Reviewer_9mm5 · 2023-11-01

**Soundness:** 3 good
**Presentation:** 3 good
**Contribution:** 3 good
**Rating:** 6
**Confidence:** 3

**Summary:**

The paper proposed an approach of defensing backdoor attack in YOLO at run-time. Specifically, the proposed LayerCAM-enabled backdoor detecotr (LeBD) utilized LayerCAM to locate the backdoor trigger, aiming to addressing the real-time requirement of application scenes in the physical world.

**Strengths:**

+ The study focuses on an interesting and important topic, the run-time defense against backdoor attacks in object detection network.
+ The paper is well-written and easy to follow.
+ The idea of using LayerCAM to locate the trigger is inspiring.

**Weaknesses:**

- The digital world and physical world

If my understanding is correct, one of the key motivations is that the existing defense focuses more on backdoor attacks in the "digital world" rather than attacks in the "physical world." However, I would suggest a more detailed and explicit definition of the digital world and the physical world. It would be better and necessary to provide a more in-depth description and explanation of why this assumption is sound. For example, you could discuss the main constraints that limit the application of backdoor attacks in the physical world. I found this assumption somehow confusing, as it suggests that a backdoored sample with a pixel-level trigger can still be printed and placed in the physical world.

- The performance of LeBD in the digital world

In Table 1, although the discussion explains that "In the physical world, affected by the shooting angle, light, and so on", photographed triggers are more vulnerable to defenses, the performance gaps of the proposed LeBD and CA-LeBD between the digital world and the physical world are still much larger than those observed in benchmarks. It appears that the performance of the proposed approach is highly influenced by the strength of backdoor attacks, with weak triggers leading to more significant performance improvements. Please provide more discussion on this point. Another concern is, the experiments in digital world scenario only involves the same backdoor attack in the physical world scenario, however, there are more attacks can be applied in the digital world, as described in previous sections.

- The application scenario of CA-LeBD

In Section 5.1, it has been claimed that "although LeBD and CA-LeBD are inferior to NEO, they are much faster than the latter". However, according to the experimental results in Section 5.4, the runtime overhead of CA-LeBD could be several times higher than NEO (if applied to all 80 classes). Please provide more discussion on how to apply CA-LeBD in practice. For instance, how to determine the appropriate number of classes when using CA-LeBD and how it might influence the defense performance.

**Questions:**

1. Please define the digital and physical worlds with a more detailed definition and explain why other backdoor attacks are hard to be applied in the physical world.
2. Why the performance gap of proposed approach is much higher than benchmarks?
3. How to determine the appropriate number of classes protected in defense when using CA-LeBD and how it might influence the defense performance?

---

> ### Author Response · Authors · 2023-11-17
> **Response to Reviewer 9mm5**
>
> We thank the reviewer for valuable time reviewing our work and for many very insightful suggestions on this work. We respond to the questions as below.
>
> * Q1: The digital world and physical world
>
> The physical world and digital world can be distinguished based on when the trigger is attached to the sample. In the digital world, an adversary first obtains a benign image and then makes some modifications to it to create a poisoned sample. In the physical world, the adversary needs to ensure that the image shot by the camera is itself the desired poisoned sample, which means the backdoor trigger needs to exist in the real world.
>
> As you say, most defenses focus on backdoor attacks in the digital world. However, most backdoor attacks in the digital world can hardly be deployed in the physical world, especially in the real-time object detection scene, because triggers of these attacks (e.g. blended[1], Poison frogs[2], PoisonInk[3], WaNet[4]) need to make pixel-by-pixel modifications to benign images. These triggers cannot be printed in advance. In contrast, using a Badnets-like pattern as the trigger is a feasible scheme in the physical world. On the one hand, the trigger can be pre-printed. On the other hand, pattern-like triggers ensure the attack successful rate because the adversary can make various transforms to the trigger during the backdoor injection phase to increase the robustness. We will add detailed explanations in the introduction of the revised version.
>
> * Q2: The performance of LeBD in the digital world
>
> There are three factors that cause the performance gap of proposed approaches much higher than benchmark. First, we set very good prior hyper-parameters for NEO as described in Appendix C. For example, the trigger blocker is larger than the trigger and the step size ensures at least 1/4 of the trigger is blocked. However, it is worth noting that these priors are not available in the actual scene. Besides, although such parameter settings have a good backdoor detection rate (TP), they also bring a high false alarm (FP). Second, although LayerCAM can roughly figure out where the trigger is, the location isn't precision all the time. The hot region in the saliency map sometimes doesn't perfectly align the trigger. LeBD utilizes the connect graph to find suspicious regions, which also increases the risk of misalignment between the occluded region and the trigger. By contrast, CA-LayerCAM locates more accurately and CA-LeBD achieves better performance. Third, the backdoor attack is robust and succeeds even if part of the trigger is occluded in the digital world, which has been extensively verified in the existing work. However, in the physical world, affected by lighting, shooting angle and photo pixel quality, the attack is naturally weakened and its robustness to trigger occlusion is reduced. So the performance in the physical world is higher than that in the digital world. It should be emphasized that the focus of this article is the backdoor defense in the physical world. At this point, our algorithms have comparable backdoor detection rates with the baseline while consume fewer time. We will add more explanation on this question in Section 5 of the revised version.
>
> In addition, you mentioned “the experiments in digital world scenario only involves the same backdoor attack in the physical world scenario”. We have to emphasize again that this article focuses on backdoor defenses in the physical world. As a result, we just consider backdoor attacks that can be deployed in the physical world. As explained in Q1, BadNets-like attacks are the most threatening.
>
> * Q3: The application scenario of CA-LeBD
>
> The first thing we need to be clear about is that as long as the source class of the backdoor attack is contained in the classes to perform CA-LayerCAM, the backdoor detection rate won't decrease. And the running time increases almost linearly with the number of classes, which is apparent from line 4-19 in Algorithm 2. To this end, we have conducted additional experiments and find that CA-LeBD is still slightly faster than NEO even in the case of 20 classes.
>
> The total classes of YOLOv5 is 80 because the network is trained on the COCO dataset. But we might just care about a few classes among them, and some classes will even never appear in a real usage scene. For example, in road monitoring, we only focus on vehicles and pedestrians rather than elephants, books and so on. Therefore, we only need to perform CA-LayerCAM on these classes. We think 5 classes are sufficient for most practical scenes.
>
> In addition, CA-LeBD is parallel for different classes, which is also a way to decrease runtime.
>
> [1] X. Chen. Targeted Backdoor Attacks on Deep Learning Systems Using Data Poisoning.
>
> [2] A. Shafahi. Poison frogs! Targetedclean-label poisoning attacks on neural networks.
>
> [3] J. Zhang. Poison Ink: Robust and Invisible Backdoor Attack.
>
> [4] Nguyen. WaNet - Imperceptible Warping-based Backdoor Attack.

---

### Meta-Review · Area_Chair_s2hu · 2023-12-08

**Metareview:**

The paper "LeBD: A Run-time Defense Against Backdoor Attack in YOLO" presents an innovative method using LayerCAM for detecting backdoor attacks in real-time in YOLO object detection systems. It is commended for its relevance, clear presentation, and consideration of real-world scenarios. However, the paper faces criticisms regarding its novelty, being perceived by some as a reimplementation of an older technique. The scope is seen as limited, and there's a lack of comprehensive experimental evaluation, particularly concerning the method's effectiveness in diverse scenarios and its ability to handle adaptive attacks. Furthermore, the paper's technical contributions are questioned, along with the specificity and broader applicability of its techniques. These concerns highlight the need for more extensive validation and discussion on the robustness and versatility of the proposed defense mechanism.

**Justification For Why Not Higher Score:**

As said in meta review.

**Justification For Why Not Lower Score:**

NA

---

### Decision · Program_Chairs · 2024-01-16

Reject